# Analysis of Yields and Their Determinants in the European Corporate Green Bond Market

Sergei Grishunin [1]  , Alesya Bukreeva [1], Svetlana Suloeva [2] and Ekaterina Burova [2,*]

1   School of Finance, Faculty of Economics, HSE University, Pokrovsky Boulevard, 11, 109028 Moscow, Russia
2   Graduate School of Industrial Economics, Institute of Industrial Management, Economics and Trade, Peter the Great St. Petersburg Polytechnic University, Polytechnicheskaya Str., 29 Academic Building #3, 195251 St. Petersburg, Russia
*   Correspondence: burova_ev@spbstu.ru

**Abstract:** The green bond market helps to mobilize financial sources toward sustainable investments. Green bonds are similar to conventional bonds but are specifically designed to raise money to finance environmental projects. The feature of green bonds is the existence of greenium, or the lower yield compared to "conventional" bonds of the same risk. The relevance of the paper is underpinned by the mixed evidence on the existence of 'greenium', especially in corporate green bond markets; there has been limited research on the topic and a narrow focus on global, US, or Chinese green bond markets. Instead, the greenium in European debt markets remains underexplored. The objective of this study is to investigate the existence of greenium and its key determinants in European corporate debt capital markets, including the local markets of the United Kingdom (UK), France, Netherlands, and Germany. The sample included 3851 corporate bonds, both green and conventional ones, between 2007 and 2021 from 33 European countries. Linear regression was applied for the analysis. The results show that the climate corporate bonds in Europe are priced at a discount to the same-risk conventional corporate bonds. The magnitude of greenium is around 3 bps. Determinants of greenium include the presence of an ESG rating and belonging to the utility and financial industry. The remaining drivers of bond yields in the European corporate debt market are the credit quality (expressed by the level of credit rating), the coupon size, the bond tenor, the market liquidity, and macroeconomic variables (growth of gross domestic product and consumer price index). For the local corporate debt markets, our results are controversial. In all markets under consideration except for the UK and the Netherlands, we did not find sustainable evidence of greenium. The results of the research lead to a better understanding of the green bond market for investors, researchers, regulators, and potential issuing companies.

**Keywords:** green bonds; greenium; European corporate green bond market; yield determinants; sustainable development; ESG



## 1. Introduction

The environmental agenda has become the most acute and significant topic of the last decade. According to The Global Risks Report (2022), the top three most severe risks on a global scale over the next 10 years are environment-related risks such as "climate action failure", "extreme weather", and "biodiversity loss". However, until 2011, environmental risks were not even included in the top five risks in terms of likelihood and impact, but economic and social risks invariably occupied the leading positions. The growing number of climate disasters prompts mankind to act and create tools to mitigate the consequences of such changes for humanity. Thus, under the Paris climate agreement, new obligations were established for countries on decarbonization-related investments. The green bond market and so-called green investment complement countries' climate change mitigation strategies. Green bonds are aimed at financing or re-financing projects helping to address

climate and environmental issues. According to Climate Bond Initiative (CBI) reports (CBI 2021b), the green bond market in 2015–2020 grew by an average of 50% per year.

Researchers and practitioners who work and investigate the green bond market have revealed the notion of negative premia to the green bonds' yield or greenium (Harrison 2022). Greenium leads to a lower yield for the buyer but allows issuers of such bonds to obtain a reduced interest rate for the issuer of green debt instruments (Bachelet et al. 2019; Ivashkovskaya and Mikhaylova 2020; Preclaw and Bakshi 2015; Zerbib 2019). The greenium is an important incentive for governmental institutions and corporates to issue more climate-related bonds.

The findings of MacAskill et al. (2021) showed that greenium was detected within 56% of primary and 70% of secondary market studies for those green bonds that are government-issued or of an investment grade. The papers which explored the nature of greenium summarized the following determinants of yield discounts: (1) investors' preferences and excess demand over supply in the green market segment; (2) liquidity level; (3) the existence of government incentives and tax breaks; (4) state of the economy; (5) macroeconomic indicators; and (6) assignment of green label and certification (MacAskill et al. 2021; Ivashkovskaya and Mikhaylova 2020; Hyun et al. 2021). Additionally, individual characteristics of the bonds and green projects of issuing companies, as well as the projects' official confirmation of compliance with Green Bond Principles or other standards, have a significant impact on the yield premium (Ivashkovskaya and Mikhaylova 2020).

However, the literature review indicates that some studies showed the absence of greenium for climate bonds in comparison to conventional ones and claimed that greenium is only a marketing tool to sell these types of debt instruments (Larcker and Watts 2020; Partridge and Medda 2020). This is an important controversy that needs to be studied further. Another controversy is that mixed results were found for corporate bond markets. Some researchers showed that, unlike green bonds issued by government arms, corporate bonds showed the absence of greenium (Bachelet et al. 2019).

Perhaps the third research gap is that most studies investigated greenium on global markets or in the largest and most liquid markets, such as the US or China. Conversely, the research coverage of the European green bond market has remained insufficient. However, the US and European green bond markets differ in many ways. Such differences cover taxonomies, the classification of green instruments, and disclosure requirements. There are also differences in the markets' liquidity, currency, credit rating structure, tenor, or deal sizes of green bonds between the US and the European debt markets.

Thus, the objective of this study is to investigate the existence of greenium and determine its key determinants in European corporate debt capital markets (including financial institutions). To achieve this objective, we will solve the following tasks: (1) to analyze the US and European corporate bond markets and find differences between them; (2) to perform a thorough review of the literature to identify the key factors determining the yield of climate bonds; (3) to study greenium and its determinants on the European corporate debt market in general and local markets of the United Kingdom (UK), France, Netherlands and Germany between 2007 and 2021; and (4) to present conclusions and discussion of the obtained results. The choice of local markets was underpinned by the large volume of corporate green bonds issued at these markets.

Our contribution to the literature is three-folded. Firstly, the paper investigates the presence of greenium and its size in European corporate green bond markets. Secondly, we analyzed the determinants of greenium in the entire market as well as in the local markets of four European countries. Thirdly, the study expands the period of analysis to 2007–2021, while earlier studies only considered the period before 2019 (Cortellini and Panetta 2021). The obtained results can be used by investors to develop strategies for managing portfolios of green bonds and by issuers of green bonds to choose the placement market. The results can also be used by other climate bond market researchers to expand the field of knowledge on this issue and identify areas for future research.

The rest of the paper is structured in the following way. Section 2 discusses green bonds and greenium and provides an analysis of corporate bond markets in the USA and Europe. Section 3 provides a thorough review of the literature on the topic. Section 4 describes the data source, the methodology, and the variables. The results of the model estimation are presented in Section 5. Section 6 contains a discussion and further research directions. Section 7 concludes the research.

## 2. The Green Bonds and the Greenium: Corporate Green Bond Markets in USA and Europe

Green bonds are fixed-income securities that are similar in financial structure to "conventional" debt (form, pricing mechanisms, market conventions, ratings, etc.) except for the differences in the usage of proceeds. The issuer of such bonds promises to use the proceeds on projects with environmental benefits (Cortellini and Panetta 2021). The green bond market has been fueled by the release of Green Bond Principles in 2014 issued by the voluntary coalitions of investors—International Capital Market Association (ICMA 2021). Moreover, another investor-focused not-for-profit international organization, Climate Bond Initiative, issued its standard (Climate Bond Standard). This document established the requirements for green bond certifications. The latter allows investors, governments, and other stakeholders to identify and prioritize low-carbon and climate-resilient investments and avoid "greenwashing" (CBI 2021a).

The establishment of these voluntary standards led to higher market integrity and set the issuance framework for green bonds. It also induced the issuance of green bonds in global and regional markets. According to CBI reports (CBI 2021b), the total size of the global green bond market exceeded USD1.6 trillion. The growth in the market is tremendous. In 2021, annual green bond issuance broke through the half trillion, ending 2021 at USD522.7 bn: a 75% increase on the prior years' volumes (CBI 2021b). The total number of issuers exceeded 2000, while the number of instruments was around 10,000. We also observed decent geographical diversification in green bond issuance. These instruments are now present in 80 countries and issued in more than 40 currencies. Recently, green bond issuances have spread to many emerging markets, particularly in China (Cortellini and Panetta 2021). However, because all these standards are voluntary, many regional green bond regulations have arisen which do not fully match each other. There are also many loopholes in local standards and regulations. It makes the green markets fragmented in terms of rules, regulations, and taxation.

The most important phenomenon of green bonds for investors and issuers is the existence of greenium. Greenium is a situation where green bonds trade at a higher price than conventional ones (thus commanding a lower yield for the buyer). The most often cited causes of greenium are that (1) companies following the "green agenda" reduce their existential risk and the risk of the ultimate default of the bond, or (2) investors are willing to sacrifice yield but meet the needs of the stakeholders (Cortellini and Panetta 2021). The question arises, however, if a negative premium is sustainable because the bonds with the same characteristics (including credit risk) should have identical yields (Bachelet et al. 2019). The answer to this question can help to project future developments in the green bond markets. This is because the greenium means a lower cost of capital from the issuers and, subsequently, a higher probability of investment opportunity (Cavallo and Valenzuela 2010). Consequently, the companies will issue more green bonds and fuel rapid growth in the green bond market. Conversely, the evaporation of greenium will halt growth. Looking ahead, based on the fact that there is no unambiguous answer to the question of whether there is a greenium, the study of this phenomenon is relevant.

The US and European green bond markets are perhaps the largest and the most established globally. The European corporate green bond market is of the most interest since around 50% of global issuances in 2021 were issued there (Figure 1).

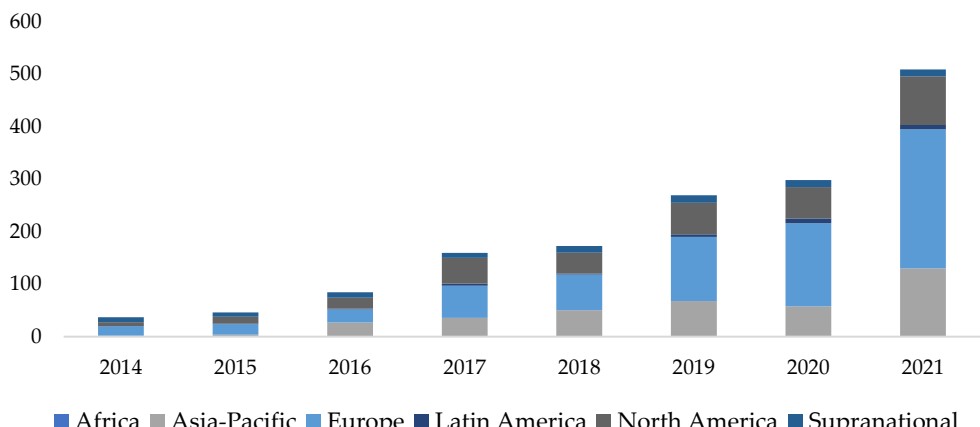

**Figure 1.** Green bond regional issuance by years (2014–2021). Source: climatebonds.net (accessed on 3 January 2023).

However, researchers exploring these markets must consider the differences between them. The distinctions are driven by differences in size, the composition of borrowers and investors, demand and supply, the credit quality of issuers, and the differences in regulatory and tax environments. For example, in the USA, the conventional corporate bond market is larger in both absolute and relative terms than the European market. The liquidity of the European Union corporate debt market is weaker due to its smaller size and the intentions of bondholders to keep the bonds until maturity. The US corporate debt market is less diverse in terms of credit ratings than the European market. This is because the US sovereign rating is AAA, while many sovereigns in Europe have lower ratings (e.g., Italy: BBB-; Portugal: BBB). Moreover, the European Central Bank's rate policy is not identical to that of the US Federal Reserve. The same is true for the key rate policy of the Bank of England (Vukovic et al. 2021).

Consequently, the US and European corporate green bonds markets differ in green bond classification systems and disclosure frameworks due to the reasons indicated above. They also differ in terms of the portrait of investors. As shown in (Deschryver and de Matiz 2020) European investors tend to be more stringent compared to their American counterparts when it comes to green investing. CBI reports show that, in time, the dynamics of demand and supply can be different and lowly correlated (CBI 2021b). There are also differences in currency, tenor, and deal sizes (Table 1). As we see from the table, the cumulative number of green bonds is higher in Europe. The average deal size is higher in Europe, while the tenor is lower than that in the US.

**Table 1.** Current state of corporate green markets in US and Europe (cumulative as for Q1 2021).

| Characteristics | US Market | European Market (Including UK) |
|---|---|---|
| Total green bonds issued (monetary), excluding Fannie Mae and supranational | USD147.5 bn | USD720 bn |
| Number of deals | 744 | 8658 |
| Number of entities | 348 | 472 |
| Average size | USD198m | EUR330 |
| Prevailing currency | USD | EUR |
| Weighted average term | 15 years | 10 years |
| Use of proceeds | Energy and water (35%) Building (45%) Transport (11%) | Energy and water (42%) Building (28%) Transport (17%) |

**Table 1.** *Cont.*

| Characteristics | US Market | European Market (Including UK) |
|---|---|---|
| Distinctive feature | Significant share of municipal bonds and mortgage-backed securities issued by Fannie Mae (62%). Bonds issued by non-financial companies comprised about 20% | Bonds issued by non-financial companies comprised about 24% |

Source: Climate bond initiative, https://www.climatebonds.net/market/data/#country-map; accessed on 3 January 2023.

These different dynamics affect both green bond prices and greenium. Additionally, greenium in markets of individual European countries is also worth studying separately. There are differences between the country's national laws or systems of national preferences and benefits for issuers and investors in green bonds (Björkholm and Lehner 2021).

## 3. Literature Review

The literature which explores the impact of the "green" status of the bond on its pricing is still emerging. Many studies use the dependent variable and the proxy of green premium to determine the difference in yields between green and conventional bonds. For example, Bachelet et al. (2019) used the difference between the ask yields of a green bond and its closest 'brown' bond neighbor in their work. Zhang et al. (2021) investigated the yield difference between a green bond and its conventional matching partner. Sheng et al. (2021) considered the issue spread between a green bond and a matched ordinary bond.

In several studies, the difference between the yields of green bonds and Treasury bonds have been shown as comparable in terms of maturity and the country in which the issue was used as a green premium (Fatica et al. 2021; Wang et al. 2019). The challenge of this approach is the difficulty in finding comparable bonds with the same maturity and cash flows (Cavallo and Valenzuela 2010). To overcome this difference, some papers define "greenium" as an option-adjusted spread OAS (Ivashkovskaya and Mikhaylova 2020; Nanayakkara and Colombage 2019, Preclaw and Bakshi 2015). For example, Nanayakkara and Colombage (2019) used the natural logarithm of OAS to analyze the difference between the daily yields of conventional bonds and green bonds on a sample of 125 bonds in 2016–2017. They found that green bonds are traded at a premium of 63 basis points against a comparable corporate bond issue. The limitation of the study is the short-time period and failure to consider bond issuance from countries with diverse financial markets. Ivashkovskaya and Mikhaylova (2020) in their paper used OAS to answer the question of whether there is 'greenium' in a sample of 2450 green and conventional bond issues from 2007 to 2020 and found a statistically significant yield premium from green bonds amounting to 23.4% in comparison to commensurable green issues. The advantage of this research was the long period (2008–2020), and the disadvantage was considering many green markets in one sample.

One more metric to measure greenium is the yield spread (YS) between the green bond yield and the risk-free rate. Wang et al. (2019) used a green bond issuance yield spread as a difference between the green bond issuance yield to maturity and risk-free rate to identify factors affecting the risk premium of China's green bond issuance. However, this approach only suits the determination of the greenium of green bonds in a single country.

The findings of MacAskill (MacAskill et al. 2021) showed that greenium was detected within 56% of primary and 70% of secondary market studies, particularly for those green bonds that are government issued, of investment grade, or that follow defined green bond governance and reporting procedures. The green premium size also depends on the green market geography and the type of the market (primary or secondary) (MacAskill et al. 2021). There are several controversies about the findings. For example, Karpf found a positive greenium on US municipal bonds (Karpf and Mandel 2018), whereas Zerbib and

Baker, drawing from moderately different methodologies and control variables, found a negative greenium (Baker et al. 2018; Zerbib 2019). Bachelet et al. (2019) argued that unlike green bonds issued by government arms or financial institutions, corporate bonds showed the absence of greenium and a lower liquidity level. The gap is underpinned by the fact that some researchers included sample green bonds of a different nature: government issued, issued by financial institutions, and non-financial companies (Ivashkovskaya and Mikhaylova 2020). Some papers showed the absence of yield discounts for green bonds in comparison to conventional ones, especially for corporate bonds, and claimed that greenium is only a marketing tool to sell these types of debt instruments (Larcker and Watts 2020; Partridge and Medda 2020). These controversies require further investigation.

The main gap between the considered studies was that they focused on the global green bond market (Ivashkovskaya and Mikhaylova 2020; Bachelet et al. 2019; Fatica et al. 2021; Loffler et al. 2021; Nanayakkara and Colombage 2019), US corporate and municipal bond market (Díaz and Escribano 2021; Karpf and Mandel 2018; Larcker and Watts 2020; Partridge and Medda 2020), or emerging bond markets (Cavallo and Valenzuela 2010; Sheng et al. 2021; Wang et al. 2019). Only a few papers addressed the greenium puzzle in the European Union (Gianfrate and Peri 2019; Agliardi and Agliardi 2021), and the outcome of these studies was mixed. The weakness of combining European and US bond markets in studies is a failure to consider the differences between those markets.

The recent literature has focused on determining drivers of greenium in addition to detecting greenium. The studies found three groups of determinants: individual bond characteristics, the financial characteristics of the issuers, and macroeconomic variables. MacAskill et al. (2021) argued that bond governance characteristics had the greatest impact on whether a green premium was evident. Investors are willing to pay a premium for bonds that offer high-quality ESG disclosures by up to 15 bps on secondary markets (Hyun et al. 2021; Baker et al. 2018). Investment grade bonds tend to provide the most predictable greenium between −2 and −6 bps (Bachelet et al. 2019; Immel et al. 2021; Sheng et al. 2021). Ivashkovskaya and Mikhaylova (2020) found that key determinants of greenium included bond durations, bond denominations, credit rating, and the amount outstanding. Other significant determinants from the individual bond characteristics group include the bond type, currency (Nanayakkara and Colombage 2019; Zhang et al. 2021), credit rating (Immel et al. 2021; Sheng et al. 2021), use of proceeds, payment rank, options (puttable/callable), and collateral type (Fatica et al. 2021; Zhang et al. 2021).

Among the financial characteristics, the following drivers were significant: revenue growth, financial leverage, ROA, ROE, ESG rating, and weighted average ESG score (Zhang et al. 2021; Immel et al. 2021). For example, Cavallo and Valenzuela (2010) highlighted the following financial drivers of greenium: firm profitability (EBIT/Assets), capitalization (Equity/Capital), firm asset size, and firm leverage (Debt/Assets). Furthermore, Febi et al. (2018) noted a significant impact of liquidity on the size of the greenium. Finally, macroeconomic factors, such as the consumer price index (CPI) or GDP growth rate, also impact the size of the greenium (Cavallo and Valenzuela 2010; Ivashkovskaya and Mikhaylova 2020; Nanayakkara and Colombage 2019). In the same way, different studies have shown different results for different markets and countries as well as types of markets (Barnett and Salomon 2012).

The methodology of much research employed an ordinary square regression (OLS) or a generalized least square regression (GLS) to time series data. The advantage of this methodology is the possibility of identifying both if the green premium is evident and what factors affect the yields in that particular market. The disadvantage of this model is that it is very difficult to match conventional and green bonds by amount, maturity, rating, coupon, or currency. Many papers used a matching approach. In this approach, bond issues are selected that are identical in terms (Loffler et al. 2021; Zerbib 2019; Gianfrate and Peri 2019; Agliardi and Agliardi 2021). The advantage of the matching approach is that it helps to isolate the greenium by controlling other characteristics of the bonds. The disadvantage of the matching approach is that it considers only the observable variables, while non-financial

factors (e.g., issuers' reputation) cannot be controlled. The other disadvantage is that it helps to discover greenium but does not allow the identification of the drivers which affect the bonds' yields.

Recently, advanced econometrics and statistical techniques have started to be employed to capture latent variables in the estimation of greenium and green bond prices. The latent variables are factors and dependencies which are not directly observable, but their influence can be seen in the measured independent variables, (Burnham et al. 1999). In cases of evaluating greenium, such latent variables may include the ESG policy of the sovereigns, distinctive features of national green taxonomies, state benefits provided to investors, issuers, and debtors, the quality of management, and corporate governance of green bond issuers, specific terms of the bonds, features of local financial markets, etc. The application of latent variable modeling techniques allow for more accurate parameter estimations and the improvement of dependent variable prediction in comparison to "conventional" regression models. This also provides insights into the interconnectedness between green bonds and the rest of the financial market. Ahelegbey et al. (2019) applied a latent factor-based classification technique to estimate a more efficient logistic model used for credit scoring. Such models can also be employed for ESG scoring. The paper argued that such a model led to an improvement in scoring performance. Other techniques include artificial neural networks (ANN), which capture latent variables. ANN could predict financial asset prices or values with higher accuracy than linear models (Tealab et al. 2017; Uma Maheswari et al. 2021). However, we found a few papers which employed these models in studying green bond prices, volume, and the greenium. We consider this a research gap. Among the existing studies, Tolliver et al. (2020) analyzed drivers of green market growth by employing structural equation modeling to study the impact of latent macroeconomic and institutional variables on green bond volume. Reboredo et al. (2020) studied the price connectedness between green bonds and financial markets using the VAR model. They found strong price spreads between green bonds markets, treasury, and foreign exchange markets. Further research on green bonds needs to consider such research tools as artificial intelligence or latent variable multivariate regression models (Burnham et al. 1999).

Let us now conclude. The analysis above shows that it has not yet been possible to accurately confirm or refute the fact of the presence of greenium, especially for green bonds issued by non-financial corporations. Conclusions about the key determinants of greenium also differ across papers. Most studies determine greenium on global, the US, or emerging (preferably Chinese) green bond markets, leaving corporate European green markets unstudied and not fully covered. We considered this fact as a weakness given that the European corporate green bond market differs from the US market in terms of size, structure, customer base, deal size, and terms. Further research directions include expanding the research tools from "conventional" linear models to more advanced techniques which can capture latent factors and the interconnectedness between factors.

## 4. Materials and Methods

### 4.1. The Model

Based on a review of the literature, a linear regression model was chosen. The model equation looks as follows:

$$Yield_{it} = \sum_i \beta_i \times X_{it} + \sum_j \beta_j \times X_j + \gamma \times a_t + \delta_i + \varepsilon_{it} \qquad (1)$$

where:

$X_{it}$—Time dependent variables.
$X_j$—Other variables (not time dependent).
$a_t$—The age of the green bond market since 2007. This metric captures the changes in the balance between supply and demand in the green bond markets across years.

$\delta_i$—Time fixed affects to consider time-varying unobservable factors that may affect the selected bond markets in a specific year.

We followed the approach of Fatica et al. (2021) and chose the current ask yield of the bond issue dependent variable ($Yield_{it}$) in our analysis. This approach agrees with that of Baker et al. (2018) and helps to accommodate green bonds from a large number of European countries with different credit qualities in a single model. The dependent variables are presented in Table 2. We chose dependent variables following the research of MacAskill et al. (2021), Ivashkovskaya and Mikhaylova (2020), Bachelet et al. (2019), and Immel et al. (2021). The variable "*Amount outstanding*" was transformed into a logarithmic form in order to decrease the scale of the data. In the Table 2, in column "expected sign", the "+" sign indicates the positive relationship between the dependent and independent variables. Conversely, the "−" sign indicates the negative relationship between the dependent and independent variables

**Table 2.** Dependent variables in the study.

| Variable Name | Expected Sign | Variable Description |
|---|---|---|
| **Time dependent variables** | | |
| Tenor$_{it}$ | "+" | The length of time until the bond is due. The more years to maturity that remain, the higher the yield spread of this bond to compensate for the growing risk. |
| Credit rating$_{it}$ | "+" | A numerical value of the top three rating agencies' (S&P, Moody's, Fitch) ratings of the bond. For modeling purposes, the ratings were translated into numeric equivalents (Appendix A). The variable is estimated as the minimum among the ratings assigned by S&P, Moody's, and Fitch Rating. A high credit rating indicates that a borrower is likely to repay the loan in its entirety without any issues, while a low credit rating suggests that the borrower might struggle to make their payments. Consequently, the lower the rating the higher should be the bond's yield. |
| ESG Rating$_{it}$ | 1/0 "−" | The dummy variable equals one if the bond has an ESG rating, otherwise, it is zero. The existence of an ESG rating of the green bond should lead to a lower yield of this bond. This is because the existence of the ESG rating indicates that an independent agency verified the purpose of the bond proceeds and evaluated the degree of integration of ESG practices into the strategy and operations of the green bond issuer. |
| Coupon | "+" | The size of the bond coupon. Higher the coupon, so should be the yield of the bond to compensate for the higher risk of such a bond in comparison to others. |
| Bid-Ask Spread | "+" | A proxy of a liquidity measure. A higher liquidity of the bond issue reflects high investors' demand which leads to the lower yield of the bond. |
| Revenue growth$_{it}$ | "−" | The measure of percentage increase (decrease) in revenue of the bond issuer over the year. The higher revenue growth indicates better economic prospects and thus the lower yield of the bond. |
| Modified duration | "+" | Modified duration measures the change in the value of a bond in response to a change in a 100-basis-point (1%) change in interest rates. The longer the duration the higher the risk of the bond, hence the higher the yield is. |

**Table 2.** *Cont.*

| Variable Name | Expected Sign | Variable Description |
|---|---|---|
| **Time dependent variables** | | |
| Debt/EBITDA | "+" | Debt/EBITDA measures an issuer's ability to repay its incurred debt. A high ratio result could indicate that a company has a too-heavy debt load, and thus, a higher credit risk. We, therefore, expect the positive relationship between this metric and the yield of the bond to compensate for the higher risk. |
| GDP Growth$_{it}$ | "−" | The growth rate of the gross domestic product in the country of the issuer's origin. The higher the GDP growth, the better the economic prospects and, consequently, the lower the yield of the bond. |
| CPI$_{it}$ | "+" | The consumer price index (CPI) also corresponds to the country of the bond's issuer. The higher the inflation, the higher the demanded yield to compensate for inflation. |
| **Non-time-dependent variables** | | |
| Green$_i$ | 1/0 "−" | A dummy variable equals one if the bond is labeled as "green", otherwise, it is zero. This variable is of the main interest in this research and the indication of greenium. |
| Industry$_i$ | 1/0 | The dummy variable equals one if the issuer operates in the particular industry (see the Data section), otherwise, it is zero. |
| **Other variables** | | |
| T | | The year of observation. |
| A$_t$ | 1–15 "+" | Reflects the number of years passed since the first issuance of green bonds. This metric captures the changes in the balance between supply and demand in the green bond markets across the years. We expect that, with the passage of time, the supply of green bonds will increase faster than the demand for those bonds. This should negatively affect the magnitude of the greenium, thus the sign of this variable is expected to be positive. |

Source: developed by authors.

However, to avoid a multicollinearity problem, we excluded variables Debt/EBITDA and modified duration from further analysis. This is because these drivers are closely correlated with the other variables. The correlation analysis is presented in Appendix B. We found a significant correlation between variables Debt/EBITDA and the credit rating as well as between variables tenor and modified duration. For the remaining variables, the variance inflation factor (VIF) was below two, and the average VIF was 1.26.

We used two methods on the basis of which it was possible to estimate the relationship between the *yield*$_{it}$ and dependent variables: fixed effect linear model and random effect linear model. To select the best model, we used the Durbin–Wu–Hausman test. In order to correct for heteroscedasticity, robust standard errors were used.

*4.2. The Data*

We followed the approach of (Ivashkovskaya and Mikhaylova 2020) and made a sample comprised corresponding issues of green and conventional corporate bonds (including bonds issued by financial institutions) from the Bloomberg database. Data sampling was carried out based on (1) the year of issue: from 2007 to 2021; (2) geographic location: Europe; (3) industry: only those industries were selected in which there were green bonds issued; (4) availability of data on the current yield and credit rating of the issue. We chose this

research period to cover the entire history of green bond issuance. Macroeconomic data was downloaded from the World Bank Database.

The initial sample contained 4035 European, both conventional and green bonds from 33 European countries for the period from 2007 to 2021. The share of green bonds in the sample was about 11%. Issues of conventional bonds with a rating not corresponding to green bond ratings were excluded from the sample. In addition, observations with an enormous amount outstanding compared to the rest of the sample, with a negative coupon, were excluded. After processing the data and removing all missing data and outliers, the sample size was 3852 bonds with a green bonds' share of about 12%. In our sample, four countries (France, Germany, Netherlands, and the UK) had the largest shares (Figure 2). We decided to consider them separately in our further analysis. The number of bonds for the UK market was 532, for France—509, for Germany—250, and for the Netherlands—342.

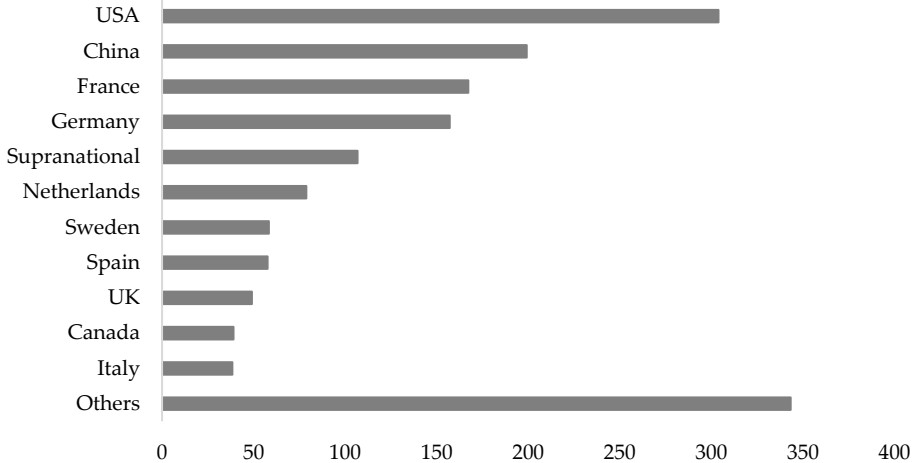

**Figure 2.** Green bond issued by country in 2021. Source: climatebonds.net, accessed on 3 January 2023.

The data was cleaned from outliers. The data of the "*Amount outstanding*" variable exceeded $3 billion, and all zero observations were excluded from the sample. The variable "*Coupon*" was cleared from outliers below zero and above 12.5%. Values of "*Yield*" above 40 points were also eliminated.

We calculated the industrial statistics of the sample to form the variable "Industry" The green issues from financial institutions accounted for around 45% of the sample, followed by utilities (30%) and industrials (10%). Thus, the green bonds from these two industries covered about 85% of the entire sample of green bonds. Therefore, we decided to form "Industry" dummies from these three industries. Table 3 presents the descriptive statistics of the final sample.

**Table 3.** Descriptive statistics of explanatory variables.

| Variables | Mean | sd | min | max |
|---|---|---|---|---|
| Yield | 2.75 | 1.92 | 0 | 25 |
| ESG Rating | 0.15 | 0.36 | 0 | 1 |
| Credit Rating | 12.97 | 4.95 | 4 | 18 |
| Green | 0.12 | 0.32 | 0 | 1 |
| Coupon | 2.85 | 1.87 | 0 | 12.50 |
| Amount Outstanding | $3.146 \times 10^8$ | $3.670 \times 10^8$ | 86.66 | $3.000 \times 10^9$ |
| Bid-Ask Spread | 0.798 | 1.55 | 0 | 63 |

**Table 3.** *Cont.*

| Variables | Mean | sd | min | max |
|---|---|---|---|---|
| Tenor | 11.43 | 8.74 | 1.51 | 100.00 |
| Revenue Growth | 11.65 | 95.88 | −94.28 | 2.34 |
| Utilities_dummy | 0.32 | 0.47 | 0 | 1 |
| GDP Growth | 0.59 | 3.55 | −10.82 | 25.18 |
| CPI | 1.31 | 0.96 | −4.48 | 7.96 |
| Fin_dummy | 0.37 | 0.48 | 0 | 1 |
| Industrials_dummy | 0.09 | 0.29 | 0 | 1 |

Source: calculated by authors.

## 5. Results

The results are divided into two parts: an analysis of the presence of a green premium in the whole European green market (based on the entire sample) and in the samples of individual countries, such as the United Kingdom (UK), France, Netherlands, and Germany since these countries have the largest samples of green bonds in our dataset.

### 5.1. Summary Outcome

Table 4 shows the results of the estimation of regression (1). The Durbin–Wu–Hausman test confirmed that the fixed effect (FE) estimator is the most efficient; thus, the study focuses on the FE estimators to explain the results.

**Table 4.** The results of model (1) estimation.

| VARIABLES | (1) Entire European Market Yield | (2) Britain Yield | (3) France Yield | (4) Netherlands Yield | (5) Germany Yield |
|---|---|---|---|---|---|
| Green | −0.0357 ** (0.0313) | −0.0278 ** (0.0263) | −0.0200 (0.422) | −0.0217 ** (0.0253) | −0.0558 (0.508) |
| ESG Rating | −0.0338 * (0.0957) | −0.107 (0.160) | −0.0512 (0.172) | −0.0349 * (0.0757) | 0.0265 (0.769) |
| Coupon | 0.976 *** (0) | 1.026 *** (0) | 0.943 *** (0) | 0.952 *** (0) | 1.010 *** (0) |
| Tenor | −0.0242 *** (0) | −0.0373 *** $(8.93 \times 10^{-6})$ | −0.0179 *** $(4.51 \times 10^{-8})$ | −0.0277 *** $(1.86 \times 10^{-5})$ | 0.00409 (0.852) |
| Utilities_dummy | −0.149 *** $(3.76 \times 10^{-5})$ | −0.190 * (0.0934) | −0.0529 (0.0158) | −0.0793 (0.162) | −0.348 (0.219) |
| Fin_dummy | −0.139 *** (0.000278) | −0.168 (0.218) | −0.0274 (0.352) | −0.0220 (0.653) | −0.238 (0.376) |
| Industrials_dummy | −0.0748 (0.143) | −0.0464 (0.816) | 0.0440 (0.531) | −0.0166 (0.775) | −0.304 (0.373) |
| Credit Rating | 0.0218 ** (0.0320) | −0.00619 (0.660) | 0.000281 (0.921) | 0.0127 * (0.0812) | 0.0167 (0.252) |
| Bid Ask Spread | 0.0295 *** (0.00382) | 0.106 (0.137) | 0.0415 *** $(3.86 \times 10^{-8})$ | 0.0663 ** (0.00295) | −0.151 (0.270) |
| Revenue Growth | −0.000121 (0.161) | $8.14 \times 10^{-5}$ (0.919) | $9.01 \times 10^{-5}$ (0.891) | −0.000289 (0.734) | −0.000293 (0.741) |
| GDP Growth | 0.00700 *** (0.00157) | 0.0133 (0.183) | 0.0127 *** (0.00174) | 0.00885 (0.249) | 0.0270 (0.324) |
| CPI | −0.0186 ** (0.0488) | −0.0250 (0.209) | −0.0416 ** (0.0335) | 0.000563 (0.980) | −0.151 (0.158) |

**Table 4.** *Cont.*

| VARIABLES | (1) Entire European Market Yield | (2) Britain Yield | (3) France Yield | (4) Netherlands Yield | (5) Germany Yield |
|---|---|---|---|---|---|
| Constant | 0.240 *** $(1.03 \times 10^{-5})$ | 0.263 (0.180) | 0.163 ** (0.0238) | 0.0821 (0.507) | 0.143 (0.597) |
| Time Passed | 0.00959 ** (0.0145) | 0.0206 ** (0.0408) | 0.0168 (0.549) | 0.0149 ** (0.0325) | 0.00416 (0.801) |
| Issuer FE R-squared | Yes 0.893 | Yes 0.856 | Yes 0.961 | Yes 0.964 | Yes 0.664 |

Robust *p*-value in parentheses: *** $p < 0.01$, ** $p < 0.05$, * $p < 0.1$.

The variables which are significant, at least at a 10% level, are highlighted in bold. All five models have a high goodness-of-fit: (1) models are significant by F-test, and (2) explanatory variables explain no less than about 60% of the yield variance.

*5.2. Results for Entire European Corporate Debt Market (33 Countries)*

Table 4 demonstrated that the green bond premium dummy variable was significant at the 5% level. Therefore, we can infer that green bonds in the entire European bond market are priced at a discount to the same risk as conventional bonds. The magnitude of greenium is around 4 bps (green bonds are priced tighter than conventional bonds). This finding matches the conclusions of most studies on the topic which confirm the existence of greenium, such as Nanayakkara and Colombage (2019), Bachelet et al. (2019), Ivashkovskaya and Mikhaylova (2020), or Preclaw and Bakshi (2015). The magnitude of greenium is close to those in other papers. For instance, Zerbib (2019) reported a negative premium of 2bps in the world bond secondary market from 2013 to 2017. Agliardi and Agliardi (2021) pointed out that many studies reported the size of the greenium in the corporate bond sector as being between 6 and 24 bps.

The variable ESG rating dummy is also significant, however, at a 10% level. It means that the greenium increases to around 7 bps if the green bond has the ESG rating. This finding coincides with that of Immel et al. (2021) that the green bonds with an ESG rating commanded a higher negative premium in comparison to unrated green bond issuances. In the same way, Hyun et al. (2021), when studying global green bond markets, reported higher greenium if the bond was certified by CBI.

Dummies belonging to the utility industry and financial industry are also significant at 1%. It means that bonds issued by utility or financial corporate can command higher greenium by 149 bps or 139 bps, respectively. This result is consistent with Gianfrate and Peri (2019) that the greenium is more pronounced for corporate issuers in the utility and power sectors. This is because these companies massively use the combination of green bonds and loans to finance projects related to renewable energy, decrease carbon emissions, and address ESG risks at their production facilities (Agliardi and Agliardi 2021). On the other hand, utility assets are significantly exposed to long-term climate change risks; thus, investing in green projects reduces the long-term default risk. Similarly, Zerbib (2019) argued that the negative premium is greater for financial bonds. However, our finding contradicts that of Fatica et al. (2021), who argued that the negative premium materializes only in favor of nonfinancial green issuers. In our opinion, financial institutions are at the forefront of the fight against climate change. Therefore, the greenium is evidence that banks need to reduce the cost of borrowing to lend more cheaply to green financing projects. Therefore, a higher greenium reflects a reduction in long-term default risks for companies financed by banks. However, the resulting contradiction needs to be studied in detail in the future.

Interestingly, the dummy of companies belonging to the industrial sector turned out to be insignificant in our research. We explain this by the fact that the concept of the industrial

sector is too broad and includes many subsectors for which the magnitude of the greenium is very different. An additional granular analysis of the greenium by industrial subsectors is needed.

As expected, the size of the coupon and the related variable of credit rating are significant at 1% and 5%, respectively. The larger the coupon size and the lower the credit rating of the bond issue, the greater the yield of the bond. Bond liquidity has a positive effect on the yield; the higher the bid-ask spread (tighter spreads usually indicate a larger volume of trading), the higher the yield. The coefficient at the variable tenor of the bond is significant at 1%. However, the sign at tenor was against our expectations. We believe that this is due to the peculiarity of the shape of the interest rate curve in the period under review.

The variable which reflects the effect of the time passed since the inception of the green bond market in the European Union is significant at 5% and has a positive sign. This result meets our expectations that, with the passage of time, the supply of green bonds increases faster than the demand for those bonds. This negatively affects the magnitude of the greenium.

Both macroeconomic variables—consumer price index (CPI) and the growth in the gross domestic product are significant at 5% and 1%, respectively. The sign at the GDP growth variable is positive, which contradicts our original expectations. We argue that GDP growth in the European economies increases demand for stocks by reducing the demand for bonds (especially in a low-interest rate environment). Moreover, a positive sign of GDP growth is supported by the finding of Cavallo and Valenzuela (2010). They argued that issuers from countries with higher GDP growth benefit from higher bond yields. This thesis is supported by macroeconomic theory as more developed countries with lower yields in the economy due to low risks for investors always demonstrate lower GDP growth and vice versa. This reduces the liquidity of bonds and the growth of yields.

An increase in CPI negatively affects bond yields. This is again contrary to our initial expectations. However, the evidence of inverse dependency between CPI and corporate bond yields coincides with the finding of Nanayakkara and Colombage (2019). Moderate inflation encourages companies' capital investments (CAPEX). The growth in CAPEX, on the one hand, drives consumption and economic growth but, on the other hand, makes fixed-income payments unattractive. The latter decreases bond yields (Nanayakkara and Colombage 2019).

Since the signs for macroeconomic variables for the entire market did not meet our expectations, we decided to additionally use the specification of the model, in which, instead of macro variables, dummies for individual countries were used. To determine for which individual countries it was necessary to introduce dummy variables, we conducted a Chow test (Appendix C). The results of the Chow test showed that dummy variables should be introduced for the UK, Denmark, Germany, and the Netherlands. The result of the estimation of the updated model is presented in Table 5.

**Table 5.** The result of model (1) estimation with dummies on countries for the entire European market.

| Variables | Entire European Market Yield | *p*-Value |
|---|---|---|
| Green | −0.0301 ** | 0.0209 |
| ESG Rating | −0.0201 ** | 0.0109 |
| Coupon | 0.972 *** | (0) |
| Tenor | −0.0243 *** | (0) |
| Utilities_dummy | −0.142 *** | $(3.43 \times 10^{-5})$ |
| Fin_dummy | −0.145 *** | (0.000140) |
| Industrials_dummy | −0.0619 | (0.204) |
| Credit Rating | 0.00556 ** | (0.0194) |

**Table 5.** *Cont.*

| Variables | Entire European Market Yield | *p*-Value |
|---|---|---|
| Bid-Ask Spread | 0.0274 ** | (0.0146) |
| Revenue Growth | −0.000104 | (0.218) |
| Country = United Kingdom | 0.0756 ** | (0.0168) |
| Country = Denmark | −0.233 | 0.176 |
| Country = Germany | 0.0764 | (0.364) |
| Country = Netherlands | 0.300 ** | (0.0168) |
| Time Passed (a) | 0.00580 | (0.146) |
| Issuer FE | Yes | |
| Constant | 0.194 *** | (0.000589) |
| R-squared | 0.894 | |

Robust *p*-value in parentheses: *** $p < 0.01$, ** $p < 0.05$.

The results of the estimation of this modified specification, in general, correspond to the results of the calculation of the original specification presented in Table 4. The green bond premium dummy variable and ESG rating variable are significant at the 5% level. In this specification, the magnitude of greenium is around 3 bps while the existence of an ESG rating increases the discount to a green bond yield of around 5 bps. Dummies belonging to the utility industry and financial industry are also significant at 1%. The magnitude of coefficients with these dummies are very close to those in the original classification and confirmed our expectations that bonds issued by utility or financial corporates can command higher greenium. As expected, "conventional" drivers of bond yields: coupon, the tenor of the bond, and credit rating, are significant. Bond liquidity has a positive effect on the yield; the higher the bid-ask spread (tighter spreads usually indicate a larger volume of trading), the higher the yield. Dummy variables for the United Kingdom and the Netherlands are significant and positive. These results coincide with 4. In these countries, greenium is positive and significant, while in other countries, greenium tends to be insignificant. The dummies of Denmark and Germany are insignificant. Interestingly, the constant is significant at the 1% level. This may be due to regulatory measures of the EU green bond market or some other drivers of supply and demand for bonds. More research is needed on this issue.

*5.3. Results for Selected European Corporate Bond Markets*

For individual European bond markets, our results are controversial. In the selected markets, we found sustainable evidence of greenium only in the UK and Netherlands. Moreover, the variables of the ESG rating dummy were insignificant in all individual markets except the Netherlands. The distinctive features of these two markets are confirmed by the outcome of the Chow test (Appendix C). Conversely, in all individual markets, the size of the coupon explained the most variance in the yields. The positive signs in the variables are in line with our expectations.

In the UK market, a green dummy was significant at a 5% level. The magnitude of greenium was around 2.7 bps (green bonds are priced tighter than conventional bonds). The utility dummy was also significant at 10%, indicating that bonds issued by a utility corporate are likely to command higher greenium by 190 bps. As expected, tenor and coupon size were significant determinants in the UK market (at a 1% level). The variable which reflects the effect of the time passed since the inception of the green bond market in the UK was significant at 5% and had a positive sign.

In France, we did not find evidence of greenium in the local green market. The significant determinants of yields in that market were the coupon; the tenor; the bid-ask spread; GDP growth; and CPI. For GDP growth and CPI, the signs were against our expectations but coincided with those of Cavallo and Valenzuela (2010) and Nanayakkara and Colombage (2019). For tenor, as for the entire EU market, the sign was against our expectations.

In the Netherlands, a green dummy was significant at a 5% level. The magnitude of greenium was around 2.2 bps. An ESG rating dummy is also significant, however, at 10%. The existence of an ESG rating increases the greenium to around 57 bps. Other important drivers of the bond yield in the Netherlands were the coupon, the tenor, and level of credit rating, the bid-ask spread, and the time factor.

In the German bond market, all variables related to the determinants of green bonds were insignificant. Interestingly, in the German market, only the size of the coupon was the significant driver of the yield. We explain our findings by the low liquidity of that bond market. It is most likely that many of the institutional investors there follow the buy-and-hold strategy. This thesis is supported by the outcome of the research of the European Commission (2017). More research is needed on this issue as well.

## 6. Discussion

The scientific novelty of our research lies primarily in the fact that we analyzed the size of the greenium and its main determinants in the European corporate bond market. An analysis of the literature showed that in most papers, global bond markets, US markets, or Chinese markets were chosen as the market under study. Conversely, the markets of state and municipal bonds were mainly studied in Europe, while the markets of green corporate issuers remained on the sidelines. Additionally, we expanded the period of analysis to 2007–2021, while earlier studies considered the period before 2019 (Cortellini and Panetta 2021).

Limitations of our study include (1) the limited number of countries included in the sample; (2) the choice of linear regression as a methodology for estimating the value of greenium; (3) the limited number of green bonds included in the sample. In particular, our sample was limited to corporate bonds only. Additionally, there was no analysis of the impact of green regulation and the benefits provided to green projects on the size of the greenium. We did not address the impact of bond labeling and verification on the magnitude of the greenium. Finally, we did not investigate the differences in the size of the greenium in the primary and secondary green bond markets of Europe.

Given the controversial result for green bond markets in individual European countries, a more detailed analysis of the yield drivers in these markets should be performed. Additional focus should also be placed on regulatory and tax incentives in selected European green bond markets as such a stimulus might be a key driver of greenium there. Furthermore, in-depth studies of individual European bond markets can deploy other methodologies, such as matching method analysis or yield curve analysis. Another question that was not addressed in our research was the difference in the sizes of greenium in primary and secondary green markets. However, some papers argue that such a difference exists (Cortellini and Panetta 2021). It is also necessary to analyze for the presence of a U-shaped dependence between the value of the greenium and its drivers and assess such a dependence (Trumpp and Guenther 2017). More work should also be conducted in the analysis of demand-side characteristics of green bonds.

If we look at the bigger picture, further research might be addressed on investigating emerging green bond markets as the number of green bond issuances has been growing there. However, existing research concentrates primarily on the Chinese market, leaving other markets behind. Such research should consider the differences in capital market development at various emerging bond markets, the differences in green bond taxonomy, carbon, and other regulation and tax issues. We also consider the analysis of the impact of the green bond issuance on the issuer's stock prices and the performance of other financial instruments as promising areas of research on the topic. This area is important as much research argues that green bonds are considered a new hedging tool against not only climate risks but also financial risks and other non-financial risks (Guo and Zhou 2021).

Lastly, the yields of green financial instruments can be determined by some latent factors such as the ESG policy of the sovereigns, distinctive features of national green taxonomies, state benefits provided to investors, issuers, and debtors, the quality of man-

agement and corporate governance of green bond issuers, specific terms of the bonds, features of local financial markets, etc. To capture these complex relationships, advanced models such as latent variable multivariate regression models or artificial intelligence tools should be used.

## 7. Conclusions

This paper is devoted to the analysis of the presence and determinants of the greenium of climate bonds in the European corporate bond market. For this purpose, we applied a linear regression model to the sample of 3852 conventional and green bonds from 33 countries from 2007 to 2021. The period under review covers the entire period of existence of green bonds in Europe since the first issue in 2007. The results showed the existence of a statistically significant negative green premium in the entire European market of around 3–3.6 bps. For the local corporate debt markets, our results are controversial. In all markets under consideration except for the UK and the Netherlands, we did not find sustainable evidence of greenium. Further research should be aimed at the impact of climate bond issuance on the issuer's stock prices and the performance of other financial instruments.

**Author Contributions:** Conceptualization, S.G. and A.B.; methodology, S.G.; software, A.B.; validation, S.G., A.B. and E.B.; formal analysis, S.S.; investigation, S.G.; resources, S.S.; data curation, A.B.; writing—original draft preparation, A.B.; writing—review and editing, S.G.; visualization, E.B.; supervision, S.S.; project administration, S.S.; funding acquisition, S.S. All authors have read and agreed to the published version of the manuscript.

**Funding:** This research was funded by Ministry of Science and Higher Education of the Russian Federation: 'Priority 2030' [(Agreement 075-15-2021-1333 dated 30 September 2021)].

**Data Availability Statement:** Publicly available datasets were analyzed in this study. This data can be found here: https://www.bloomberg.com/professional/solution/bloomberg-terminal/ (accessed on 3 January 2023).

**Conflicts of Interest:** The authors declare no conflict of interest.

## Appendix A

**Table A1.** Transformation of rating scales.

| Fitch | Moody's | S&P | Category |
|---|---|---|---|
| AAA | Aaa | AAA | 1 |
| AA+ | Aa1 | AA+ | 2 |
| AA | Aa2 | AA | 3 |
| AA− | Aa3 | AA− | 4 |
| A+ | A1 | A+ | 5 |
| A | A2 | A | 6 |
| A− | A3 | A− | 7 |
| BBB+ | Baa3 | BBB+ | 8 |
| BBB | Baa2 | BBB | 9 |
| BBB− | Baa1 | BBB− | 10 |
| BB+ | Ba3 | BB+ | 11 |
| BB | Ba2 | BB | 12 |
| BB− | Ba1 | BB− | 13 |
| B+ | B3 | B+ | 14 |
| B | B2 | B | 15 |
| B− | B1 | B− | 16 |
| CCC+ | Caa3 | CCC+ | 17 |
| CCC | Caa2 | CCC | 18 |
| WD | NR | NR | 19 |

## Appendix B

**Table A2.** Correlation matrix.

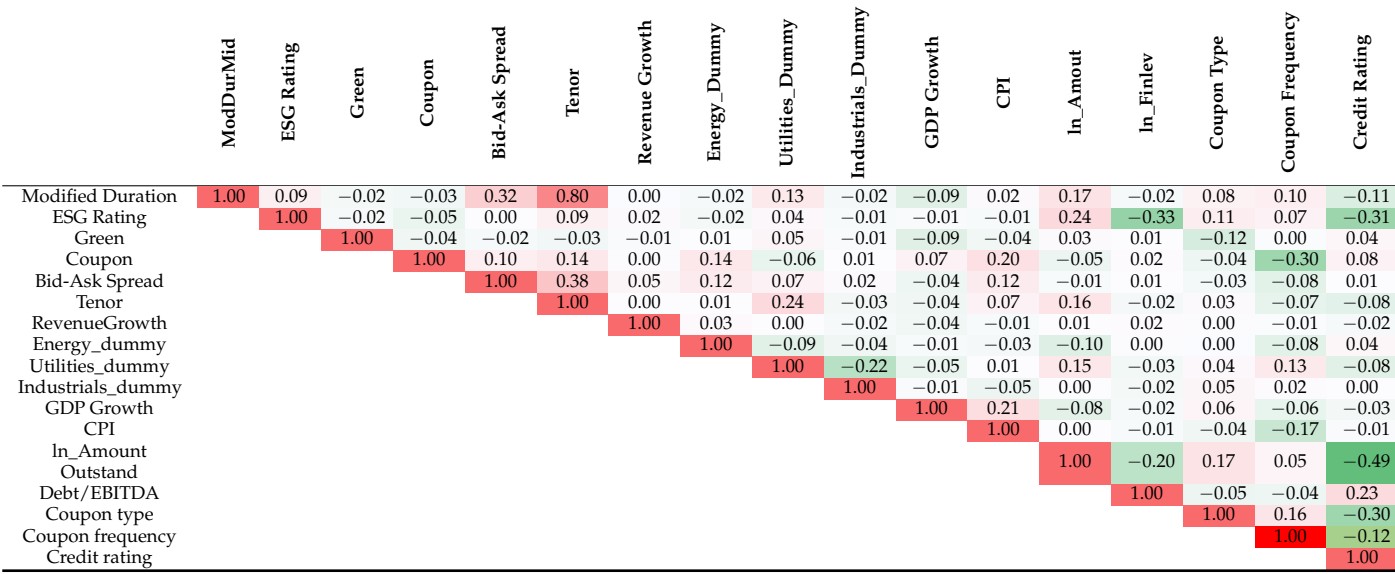

Note: the color indicates the strength of correlation between factors. Green indicates acceptable level of correlations between variables, red indicates the strong correlation between variables which indicates concerns. Shades of color mean different strength of correlation between variables, the more intense the color, the stronger the degree of connection between variables.

## Appendix C

**Table A3.** Estimation of Chow test.

| Country | F-Statistics | *p*-Value | Include in the Model? |
| --- | --- | --- | --- |
| Austria | 0.36 | 0.969 | No |
| Belgium | 0.65 | 0.762 | No |
| UK | 2.47 | 0.004 | Yes |
| Bulgaria | 0.2 | 0.94 | No |
| Croatia | 0 | 0.99 | No |
| Czech | 0.33 | 0.97 | No |
| Denmark | 2.16 | 0.02 | Yes |
| Estonia | 0.06 | 0.99 | No |
| Finland | 0.24 | 0.99 | No |
| France | 1.52 | 0.114 | No |
| Germany | 3.8 | 0 | Yes |
| Greece | 0.02 | 0.99 | No |
| Hungary | 0.19 | 0.99 | No |
| Iceland | 0.15 | 0.86 | No |
| Ireland | 1.57 | 0.107 | No |
| Italy | 0.35 | 0.98 | No |
| Jersey | 0.44 | 0.5 | No |
| Lithuania | 0 | 0.99 | No |
| Luxembourg | 0.75 | 0.69 | No |
| Malta | 1.01 | 0.43 | No |
| Netherlands | 24.5 | 0.0 | Yes |
| Norway | 0.68 | 0.75 | No |
| Poland | 0.55 | 0.85 | No |
| Portugal | 0.41 | 0.94 | No |
| Romania | 0.01 | 0.93 | No |
| Slovakia | 0.5 | 0.86 | No |
| Spain | 0.16 | 0.99 | No |
| Sweden | 1.55 | 0.107 | No |
| Switzerland | 0.48 | 0.92 | No |

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
