# Peer review of "Analysis of Yields and Their Determinants in the European Corporate Green Bond Market"

_risks, doi:10.3390/risks11010014_

Round 1

Reviewer 1 Report

This article analyzes the presence, direction and determinants of the risk premium in the European green bond market using Bloomberg data on a sample of 3852   conventional and green bonds from 33 European countries for the period from 2007 to 2020. Using a multiple OLS regression model, the paper shows the existence of a positive green premium in European countries bond market (except Germany) driven by ESG  and credit rating of the issuer, its financial performance, as well as macroeconomic factors (CPI and GDP growth rate).

The paper should make it clearer what is the positioning and contribution to the existing literature , eg Ehlers and Packer (Bank Int. Settlements Q.Rev 2017)  Karpf and Mandel (Nat. Clim. Change 2018) Hachenberg and Schiereck (J. Asset Manage. 2018)   Zerbib (JBF 2019) . Also what is really new compared to  Ivaskovskaya and Mikhailova (Jour. Corp. Fin research 2020) ?

The methodology is not discussed, for instance why not applying a DID and PSM methodology rather than only OLS regressions ? It should be much more convincing that the application of such a methodology in this context is sound, which presently is very difficult to guarantee. It would be necessary also to test the robustness of such an approach

Author Response

Thank you very much for your comments.

We re-write the introduction to add the scientific novelty and more detailed reasoning for the paper. We also added the literature review section which answer the question what is the positioning and contribution of the paper to the existing literature. Also, we added the explanation of the choice of the methodology. 

Reviewer 2 Report

Referee report

Paper: In Search of Greenium. Empirical Analysis of Risk Premiums in the European Green Bond Market

The paper aims to investigate the existence of a difference in yield between green bonds and conventional or brown bonds. The topic is of relevant interest as sustainability issues have gained considerable importance among investors, companies, and regulators.

However, I have major concerns over the paper.

1.     Theoretical contribution and motivation

The contribution of the paper is limited in scope. Just “to investigate the presence and direction of the risk premium in the European green bond market” is not an important novelty per se. Authors should provide more solid and convincing explanations whether European issuers displays some traits that differ them from other countries’ issuers.  For example, it can be detected if there are different drivers of the risk premium for European issuers. The topic must be investigated more deeply.

Along this line, the introduction does not properly outline the contributions of the paper.

A review of the literature is missing. Authors should consider adding such a paragraph.

2.     Sample

The stated purpose of the authors is to investigate the risk premium in the European bond market. Hence the reader expects that all issuers on the European bond market (or all European issuers?) should be included in the sample. Then why focus only on corporate issuers, as clearly stated only at line 204? This is confusing and should be clarified.

3.     Methodology

How can it be detected if the green bond has a different yield than a conventional bond, i.e. a bond that has the same characteristics except the attribute of “greenness”?

It is customary in the literature to use a matching method and form a sample of green and “comparable” conventional bonds. Then the “greenium” is defined as the difference between the yield on a green bond and the yield on the matched conventional bond.

I would suggest that the authors follow a matching procedure. It grants soundness and helps to resolve the problem of heterogeneity among bonds.

Second issue is whether to measure the greenium at issuance (primary market) or in the secondary market. You state that “…availability of data on the current yield and credit rating of the issue. All data are on the date of 02/22/2022”. If I get it right (but there is no other information in your paper), this means that you are measuring the yield on exactly that date. This does not make sense. What conclusions can be inferred from a single point in time?

Besides, I do not see the point of focusing on single-country data. Are there any country-specific variables that can explain the differences in the results? This should be investigated.

In the regressions in Table 2 I do not understand why some dependent variables are missing when regressed on a single country. For example, your main explanatory variable (green dummy) is missing for Germany.

Author Response

Thank you very much for your valuable comments they really helped us to make the paper better.

  1. We added the literature review section. In this section we put the the part 2.4 where we explain sexplanations whether European issuers displays some traits that differ them from other countries’ issuers
  2. We re-write the introduction to stress the scientific novelty and practical significance
  3. The sample section was rewritten to clarify the choice of the bonds
  4. Here we still decided to leave OLS regression but in the conclusion we stressed that matching sampling will be used in our next study. However, to achieve the objective of this particular paper the OLS is best suit. We added in the text the explanation why (see section 2.3 in literature review)
  5. Also, we changed the wording in section 3.1 to correct the mistake …availability of data on the current yield and credit rating of the issue. "all data are on the date of 02/22/2022” (this was defintely wrong)
  6. We also re-write the results section to avoid the sitatution "why some dependent variables are missing when regressed on a single country"

Reviewer 3 Report

risks-1759297 “In Search of Greenium. Empirical Analysis of Risk Premiums in the European Green Bond Market”

 Thank you for the opportunity to read your manuscript.

This manuscript certainly addresses a relevant research question and of potential interest to Risks’ audience. The manuscript is well written, easy to follow, and the empirical design seems to reflect relevant and related literature on the topic. However, both the research gap and the empirical design need some improvements to prove the manuscript additional contribution to the related literature.

1.     The topic is relevant and contemporaneous, but the research gap is not clear or convincing. Arguing that previous studies looked mainly at US and Chinese contexts, is not enough to motivate new research on the topic. The authors should better identify the research gap to motivate this study. One way to do this is to explore the contextual differences between US or China and European countries, in terms of capital markets (e.g. the cross country differences in shareholder protection and creditor rights) and relate these issues to financing decisions and risk premiums.

2.     Another concern is the fit between the use of panel data and model specification and estimation method. It is not clear to me whether and how this issue has been addressed along the analysis. I believe these methodological aspects should be addressed and/or clarified.

3.     The European context is a strength of the analysis and includes various countries. Thus, the authors should provide more information about the countries in the sample, for example, who they are, they representativeness in the whole sample, the proportion of GB and CB per country.

4.     The measurement of all variables is described and supported by previous literature, but the source of the data is not always clear. More information about the data source should be provided.

5.     The period (2007 – 2020) is a strength but needs to be better justified and the benefits better explained given that some economic shocks happened along the (long) period.

6.     I wonder if the authors can provide a graphic with the evolution of GB and CB in this market along the period of the investigation.

7.     It is not clear to me why in Table 2 the set of independent variables are not identical among the models. Please, provide more details.

8.     Given the purpose of GB, and previous studies on this issue, I have the impression that this is an industry sensitive issue (like ESG performance). Thus, I am surprised by the simple industry related variable (Utilities_dummy) used in the empirical analysis. I wonder if the authors can better justify this choice.

9.     Related to the above comment, and more important, better exploring this issue (industry sensitivity of risk premium of GB) can be an incremental contribution of your study in comparison to the literature you cite.

10.  Some robustness tests with regards sample composition and measurement of dependent variables would strengthen the validity of the analysis and the conclusions.

 Good luck forward and I hope the authors will find my comments useful to improve this manuscript.

Author Response

Thank you very much for your review. It was very valuable and helped us to improve the paper

  1. We added the literature review section and their in section 2.4 we explained the motivation for choosing European debt markets. We also rewrite introduction to stress scientific novelty and practical significance.
  2. Methodological choice also was clarified in section 3
  3. Information about countries also was added to section 3.1
  4. The sources of variables was clarified in section 2.2
  5. The period between 2007-2021 was justified in section 3.
  6. We also rewrite the result section (section 4) to answer the concerns 6-10

Reviewer 4 Report

The authors of the research paper “In Search of Greenium. Empirical Analysis of Risk Premiums in the European Green Bond Market” present a topic very relevant to green bond market.

First of all, I would like to point out that method and empirical results have been properly realized and the research methodology is appropriate.

My main concerns about the paper are two.

First, I cannot see clear enough what the main goal is, probably because the introduction is very short and fails in the attempt to give us a framework for the paper. I do suggest the authors to rewrite it. Also, probably conclusions should be extended in order to provide with a more detailed final idea.

Secondly, and probably the weakest part of the manuscript is the literature review.

I do suggest to the authors that a review of very recent literature about the subject, specifically from wider international point of view should be carried out. In fact, there are some recent and very relevant publications on this topic that should be included. Let me suggest some very recently references on this topic that should be included on your paper:

Wang, J.; Chen, X.; Li, X.; Yu, J.; Zhong, R. The market reaction to green bond issuance: Evidence from China. Pac. Basin Financ. J.2020, 60, 101294; also, Lebelle, M.; Jarjir, S.L.; Sassi, S. Corporate Green Bond Issuances: International evidence. J. Risk Financ. Manag. 2020,13, 25; also, Bernabé Argandoña, L.C.; Cruz Rambaud, S.; López Pascual, J. The Impact of Sustainable Bond Issuances in the Economic Growth of the Latin American and Caribbean Countries. Sustainability 2022, 14, 4693. https://doi.org/10.3390/su14084693;  also, Cheong, C.; Choi, J. Green bonds: A survey. J. Deriv. Quant. Stud. 2020, 28, 175–179; and finally, Dan, A.; Tiron-Tudor, A. The determinants of green bond issuance in the European Union. J. Risk Financ. Manag. 2021,14, 446

Author Response

Dear Sir or Madam. Thank you very much for your valuable comments:

  1. We rewrite the intro section to clearly show the goal and scientific novelty of the paper
  2. We added the detailed literatire review section (Section 2) which we hope addressed all the issues of the reviewer

Round 2

Reviewer 2 Report

The authors have made improvements with respect to previous version. However, not all suggestions have been properly addressed and I still have major concerns.

1.      Methodology

What is greenium? This neologism reflects the difference in the yield of green bonds with respect to (similar) brown bonds.

In order to measure it, authors rely on yield spread (YS) (i.e., the difference between the yield on a bond and a riskless rate). First, this approach is marginal, as was previously used in literature by Wang et al. (2019) only. 

Second, I disagree that this measure is “the most convenient measure of bond risk premia in terms of data availability, analysis simplicity and clarity of interpretation” (line 127). To me it adds confusion.

In Wang’s paper, the purpose of the analysis was to analyze the determinants of the spread on a sample of green bonds in a single country (China).

The authors build a sample of European bonds issued in 33 European countries. As a proxy for the risk- free rate, they use the ECB risk-free rate, i.e., a risk-free rate (denominated in euros? But you also have non-euro countries in your sample. How do you deal with that?) for AAA debt (reasonably, Germany). But as you correctly state, in your sample there are countries non-AAA rated. So, your YS also reflects country risk.

If you want to preserve your model, I suggest that instead you follow Fatica et al. (2021) and use just the bond yield as your dependent variable.

Also, in your model you should add year fixed effects as well as firm fixed effects (instead of industry) in order to control for time-invariant factors.

2.      Sample and variables

It takes quite a bit to the reader to understand that the focus of the paper is on corporate bonds. This should the clearly stated right from the beginning and the writing of the paper should be less generic and more focused on the corporate issue. Finally, the title of the paper should be adjusted accordingly.  

In line 83, it is stated that the sample period is 2007-2021, whereas, in line 92, authors refer to 2019-2021. This should be checked.

In line 348, authors state “Bid-Ask Spread is a proxy of a liquidity measure which is calculated as the difference between ask and bid price for this issue as of February 2022”. I do not understand as of February 2022. Are you considering bid-ask at the issuance date of each bond in the sample?

Revenue growth, GDP Growth and CPI: authors do not provide any explanation of the predicted sign (line 361-366)

Table 5. what’s the meaning? I do not see the rationale for re-running the model without variables that are not significant.

3.      Literature review

What is found in the literature review’s section is more a paragraph on methodological issues than a proper literature review. The authors should address this and restate the section according to standard practice.

Par. 2.4 is not clear to me. I do not understand the point of comparing US and EU markets. If the purpose is to state that there are different results according to different markets, than this issue can be properly accounted for in a “standard” literature review.  Table 1 negatively affects clarity and scope of the paper.

4.      Discussion

The discussion section still deserves great improvement. The explanations the authors provide are generic statements, not related to hypothesis eventually developed in literature review’s section.

Author Response

Dear Sir or Madam,

Thank you very much for your valuable comments. We are absolutely sure that your suggestion will help us to develop further

  1. We changed the approach to Fatica et al and fully recalculated all the models (as suggested)
  2. We corrected mistakes in sample and variables
  3. We rewrite almost all the paper, inlcuding parts results and discussion

Reviewer 3 Report

The manuscript has made some obvious improvements since the first version. I commend the authors for being responsive to previous comments.

Although the research question is now better motivated and the literature review better connected with the study purpose, the structure and clarity in the contributions are my concerns.

More specifically, the way the literature review is structured is like a Research Methods section, where the authors justify methodological choices. I strongly recommend the authors to reassess the structure and writing, in order the better fit the literature review with journal standards. A literature review section should emphasize previous findings (not measures and estimation methods), research gaps and how the current study fill up the gap in prior research.

The discussion and conclusion section should be expanded. How the results explain risk premiums, and why its determinants are different among European “local markets” should be better explored, more in depth discussion.

The authors should better discuss why the “European” market presents so many differences with respect to the object of this study. If local markets are so different, why at the front end the authors treat the European market as monolithic, homogeneous. The evidence on such differences is a contribution of the study that the authors did not explore enough along the discussion.

Finally, the discussion and conclusion sections should come back to the main goals stated in the introduction and the literature used. I mean the authors should better explain how the evidence provided complement this literature, how the period used finally provided better picture of the issue, what are exactly the implication of their findings for investors.

Author Response

  1. We changed the structure of the article as per journal template (as suggested)
  2. We fully rewrite results and discussion sections to follow the comments provided

Thank you vey much for your valuable comments. They really help.

Round 3

Round 4

Reviewer 2 Report

Authors have incorporated suggestions and comments in their revision

Author Response

Thank you very much for your valuable comments and support.